# Radiological Evaluation of Combined Anteroposterior Fusion with Vertebral Body Replacement Using a Minimally Invasive Lateral Approach for Osteoporotic Vertebral Fractures: Verification of Optimal Surgical Procedure

**DOI:** 10.3390/jcm11030629

**Published:** 2022-01-26

**Authors:** Takumi Takeuchi, Kenichiro Yamagishi, Kazumasa Konishi, Hideto Sano, Masato Takahashi, Shoichi Ichimura, Hitoshi Kono, Masaichi Hasegawa, Naobumi Hosogane

**Affiliations:** 1Department of Orthopedic Surgery, Kyorin University, Tokyo 181-8611, Japan; kimotako1107@yahoo.co.jp (T.T.); koniayu@hotmail.co.jp (K.K.); hidetosano@ks.kyorin-u.ac.jp (H.S.); mtaka@ks.kyorin-u.ac.jp (M.T.); ichimura@ks.kyorin-u.ac.jp (S.I.); 2Department of Orthopedic Surgery, Higashiyamato Hospital, Tokyo 207-0014, Japan; yamakens2000@yahoo.co.jp; 3Department of Orthopedic Surgery, Keiyu Orthopedic Hospital, Tatebayashi 374-0013, Japan; kouno-h@ku-kai.or.jp; 4Department of Orthopedic Surgery, Kugayama Hospital, Tokyo 157-0061, Japan; rabiopapa@gmail.com

**Keywords:** minimally invasive spinal treatment (MIST), osteoporotic vertebral body fracture (OVF), combined anteroposterior fusion, vertebral body replacement (VBR), wide footplate expandable cage, percutaneous pedicle screws (PPS)

## Abstract

The combined anteroposterior fusion with vertebral body replacement (VBR) using a wide footplate expandable cage with a minimally invasive lateral approach has been widely used for pseudoarthrosis after osteoporotic vertebral fractures. The purpose of this study is to evaluate the radiological results of combined anteroposterior surgery using VBR and to recommend the optimal procedure. Thirty-eight elderly patients were included in this study. The mean preoperative local kyphosis angle was 29.3°, and the mean correction loss angle was 6.3°. Cage subsidence was observed in ten patients (26.3%), and UIV or LIV fracture in twelve patients (31.6%). Patients with cage subsidence were compared to those without cage subsidence to determine the causal factors. The mean number of fixed vertebrae was 5.4 vertebrae with cage subsidence and 7.4 vertebrae without cage subsidence. In addition, to precisely clarify the optimal number of fixed vertebrae, those patients with two above–two below fixation were compared to those with less than two above–two below fixation, which revealed that the correction loss angle was significantly less in two above–two below fixation (*p* = 0.016). Based on these results, we recommend at least two above–two below fixation with VBR to minimize the correction loss angle and prevent cage subsidence.

## 1. Introduction

The number of patients with osteoporotic vertebral fractures (OVF) is increasing due to the aging population in Japan, and orthopedic surgeons are treating an increasing number of such patients on a daily basis. While the importance of treatment for OVF has been reported [1,2], pseudoarthrosis and kyphotic deformity after OVF often create a number of problems which can make treatment difficult [3]. Balloon kyphoplasty (BKP) is indicated for patients with a mild collapse of the vertebral body [4,5], but in patients with neurological deficit or sagittal imbalance associated with severe vertebral body collapse, reconstruction of the anterior column is required, which can be difficult due to bone fragility [6].

Recently, there have been several multicenter studies on the surgical treatment for OVF in Japan [3,7,8], and many studies have reported various surgical techniques for this condition, such as anterior spinal fusion (ASF) [7,9,10,11], posterior spinal fusion (PSF) combined with vertebroplasty (VP) [12,13,14,15,16,17,18], PSF with 3-column osteotomy (3-CO) [19,20,21], and combined anteroposterior fusion [17,21,22].

Some reports have recommended the augmentation with additional posterior fusion in OVF patients undergoing ASF alone to prevent kyphotic deformity progression or screw loosening due to poorer stability of the anterior vertebral screw [11,16]. PSF with VP is widely used for OVF patients because it is a less invasive procedure with a lower perioperative complication rate [18], but one of the disadvantages of this technique is the high incidence of correction loss [15]. Combined anteroposterior fusion provides optimal biomechanical constructs with an anterior strut graft and posterior fixation and has an advantage in kyphosis correction through the anterior approach at the OVF site [17]. However, this technique is highly invasive for elderly patients, with higher intraoperative blood loss [23] and a longer surgical time. PSF with 3-CO has advantages in kyphosis correction and direct decompression by using the posterior approach alone. However, 3-CO is also a highly invasive surgery with massive blood loss, and it has a risk of neural tissue injury [19,20,21]. All of these reported techniques have some disadvantages, as described above. Therefore, the surgical strategy to obtain sufficient stability with less invasion should be established for the treatment of OVF, especially in elderly patients with poor bone quality.

Recently, the widespread use of minimally invasive spinal treatment (MIST) has made it possible to perform combined anteroposterior fusion with vertebral body replacement (VBR) using a wide footplate expandable cage with a minimally invasive lateral approach [24,25]. The anterior surgical approach can be performed in the same way as the minimally invasive lumbar-lateral interbody fusion (L-LIF) approach, and corpectomy can achieve a fixation with less adjacent tissue damage [26]. However, as this technique is relatively new, there is a lack of sufficient evidence to determine the optimal procedure, such as the surgical sequence or the range of fixation. The purpose of this study is to evaluate the radiological results of combined anteroposterior fusion using VBR and to investigate its optimal surgical procedure.

## 2. Materials and Methods

### 2.1. Subjects of the Study

Patients who had undergone combined anteroposterior fusion using a wide footplate expandable cage system with a minimum one-year follow-up at our four hospitals were included in this study, and those with augmentation such as with an ultra-high molecular weight polyethylene cable, and patients without the required radiographs, were excluded.

A total of 38 patients (17 males and 21 females) with a mean age of 75.2 ± 7.4 (53–85) years and a mean follow-up of 21.2 ± 10.4 (12–51) months were enrolled. The affected vertebra was at the thoracolumbar junction level (T11-L2) in 27 patients (71.1%) and the lumbar level (L3-4) in 11 patients (28.9%). Twenty-eight patients had highly unstable pseudoarthrosis after OVF, and ten patients had kyphotic deformity after OVF. The sequence of surgery was anterior surgery followed by posterior surgery (A-P surgery) in 21 patients (55.3%), and posterior surgery followed by anterior surgery (P-A surgery) in 17 patients (44.7%) (Table 1). X-Core2^®^ (NuVasive, San Diego, CA, USA) was used in 35 patients, and T2 Stratosphere ^TM^ (Medtronic, Memphis, TN, USA) was used in 3 patients.

### 2.2. Surgical Procedure

In the patients whose kyphosis improved in the supine position, P-A surgery was performed. In contrast, those patients with rigid kyphosis due to bony fusion between the fractured vertebra and its adjacent vertebra or those patients with OVF at the lower lumbar spine underwent A-P surgery. The anterior surgical approach was performed in the same way as the lumbar-lateral interbody fusion (L-LIF) approach, with the retroperitoneal approach at the mid–lower lumbar spine level, and with the extra-pleural approach at the thoracolumbar spine transition level combined with the trans-diaphragm approach as needed. For corpectomy, the discs above and below the affected vertebrae were thoroughly removed, and the fractured vertebra was resected piece-by-piece after ligating the segmental arteries. Posterior surgery was performed using percutaneous pedicle screws (PPS) in 16 patients without neurological deficits, and the other 22 patients were fused with the conventional open technique combined with the decompression of neural tissues. In 28 patients whose symptoms were caused by localized pseudoarthrosis, fixation was performed at the affected vertebra, including several vertebrae above and below, with the range of fixation determined by the surgeons involved. One above–one below fusion was performed in the early phase of VBR introduction, and thereafter the range of fixation was two above-two below or more in the majority of the patients. In 10 patients requiring kyphosis correction, fixation from the ilium to the lower thoracic spine was performed using the same strategy as in adult spinal deformity surgery [27]. The patients were allowed to leave their beds using a hard brace a few days after the surgery. The hard brace was used for a minimum of 6 months after the surgery. All surgeries were performed by a board-certified spine surgeon (approved by the Board of the Japanese Society for Spine Surgery and Related Research).

### 2.3. Measurement Parameters

Radiographs were taken before and immediately after the surgery, and at the final follow-up in the standing position for those who could stand, and in the sitting or supine position for those who could not stand. The radiological evaluation included the local kyphotic angle (LKA), which was defined as the angle between the upper endplate of the proximal adjacent vertebra and the lower endplate of the caudal adjacent vertebra, and the correction loss angle, which was defined as the difference in the LKA immediately after surgery and at the final follow-up. In addition, the definition of bone fusion was the presence of bone bridging between a fractured vertebra and its adjacent vertebra, which was confirmed with the coronal or sagittal view of computed tomography (CT).

Mechanical failures were classified as follows: (1) 3 mm or more endplate injury by immediate postoperative X-ray or CT, (2) cage flotation, which was the gap between the cage and the endplate by immediate postoperative X-ray or CT, (3) 3 mm or more cage subsidence at the final follow-up X-ray or CT, (4) fractures of the upper instrumented vertebra (UIV) or the lower instrumented vertebra (LIV), (5) adjacent vertebral body fracture (AVF), and (6) loosening of the pedicle screws (PS).

The clinical evaluations included perioperative complications, surgical site infection, and ability of daily living (ADL) at the baseline and at the final follow-up, which were classified into 4 stages: stage 1: able to walk independently, stage 2: able to walk with assistance, stage 3: able to sit independently, and stage 4: unable to sit unaided [22].

For statistical analysis, comparisons between the two groups were conducted using the *t*-test (normality) and the Mann-Whitney U test (non-normality), and the χ^2^ test or the Fisher’s exact test were used for the variables.

## 3. Results

The patients’ demographics and surgical outcomes are listed in Table 1. The mean preoperative LKA was 29.3° (−21–62), which was corrected to −0.3° (−24–23) after the surgery with a correction angle of 29.6° (−1–54). LKA was 4.6° (−26–41) at the final follow-up with a mean correction loss angle of 4.9° (−15–20). Mechanical failures occurred in 26 patients (68.4%), some of whom showed more than one failure, out of which were cage subsidence in 10 patients (26.3%), UIV or LIV fracture in 12 patients (31.6%), AVF in 2 patients (5.3%), intraoperative endplate injury in 3 patients (7.9%), cage floatation in 3 patients (7.9%), and PS loosening in 8 patients (21.1%) (Table 1). All 38 patients had bone fusion at the vertebral body replacement site evaluated by CT at the final follow-up.

As cage subsidence was observed in a quarter of patients, the factors related to this were further evaluated by comparing the patient groups with and without cage subsidence. As a result, the mean number of fixed vertebrae was 5.4 ± 4.5 in the group with cage subsidence and 7.4 ± 3.6 in the group without cage subsidence. The number of fixed vertebrae was significantly less in the cage subsidence group (*p* = 0.003) (Table 2).

In addition, there was a weak but apparent negative correlation (R = −0.37) between the number of fixed vertebrae and the correction loss angle (*p* = 0.023) (Figure 1). There were no significant differences in the patients’ demographics or the surgical sequence.

There was a weak but apparent negative correlation (R = −0.37) between the number of fixed vertebrae and the correction loss angle (*p* = 0.023).

The range of fixation in the cage subsidence group was one above–one below in six patients, and two above–one below in two patients. In the latter patients, the cage subsidence occurred at the caudal level with one vertebra fixation in both patients. Although two patients with long-range fixation had cage subsidence, these patients had Parkinson’s disease (PD). Therefore, we hypothesized that at least two above-two below fixation, which is hereafter referred to as the five vertebrae fixation group which includes the affected vertebra (5VG), is a prerequisite to prevent cage subsidence. To confirm this hypothesis, 5VG was compared with the one above–one below and the two above–one below fixation groups, which are hereafter referred to as the four or less vertebrae fixation group including the affected vertebra (4LVG). The correction loss angle was 4.8° ± 5.0° in 5VG, which was significantly smaller than that of 4LVG which was 12.3° ± 7.8° (*p* = 0.016), and no cage subsidence was observed in 5VG (*p* = 0.0001). The 5VG consisted of more female patients and those who had predominantly P-A surgery with a shorter follow-up period. There were no significant differences in age (*p* = 0.39) or in UIV or LIV fracture (*p* = 0.21) or in AVF (*p* = 0.55) (Table 3).

Furthermore, since residual kyphosis due to under-correction was suspected as one of the factors related to correction loss or cage subsidence in 5VG and 4LVG patients, the comparison was made between the patients with satisfactory correction (postoperative LKA < 5°) and patients with inadequate correction (postoperative LKA ≥ 5°). There were no significant differences in correction loss angle (*p* = 0.299), cage subsidence (*p* = 0.246), or in the sequence of the surgery (*p* = 0.639) (Table 4).

The other noticeable mechanical failures were cage flotation (3 patients) and endplate injury (3 patients). Five of these six patients had A-P surgery, suggesting that A-P surgery may pose a risk for cage malposition (Table 1).

Perioperative complications included two patients with neurological deficits, one patient with postoperative delirium, one patient with postoperative hemothorax, and one patient with urinary tract infection. There was no surgical site infection during the follow-up period.

Figure 2 shows the ADL stage at the baseline and at the final follow-up. At the final follow-up, 20 patients improved by 1 stage, 8 patients improved by 2 stages, 8 patients improved by 3 stages, and 2 patients remained unchanged. There was no patient whose ADL stage worsened. At the final follow-up, the results showed that 33 patients were able to walk independently (stage 1), 3 patients were able to walk with assistance (stage 2), and 2 patients were able to sit independently (stage 3) (Figure 2).

## 4. Illustrative Cases

### 4.1. Case 1

A sixty-nine-year-old male had severe lower back pain due to pseudoarthrosis at L4 after OVF. VBR at L4 was performed followed by short-segment fixation (one above-one below) at L3-5 with PPS without decompression. The surgical time was 196 min, and the blood loss was 110 g. No complications occurred during the perioperative period. Cage subsidence occurred gradually but bone fusion was observed at six months after surgery. The local kyphosis angle improved from 24° before surgery to −21° immediately after the surgery and to −11° at the final follow-up (Figure 3).

### 4.2. Case 2

A seventy-five-year-old male had severe lower back pain and right lower leg pain, due to pseudoarthrosis at L4 after OVF. VBR was performed at L4 followed by two above–two below PLF at L2-S1 combined with L-LIF at L2/3 and L5/S1 posterior lumbar interbody fusion (PLIF). The surgical time was 413 min, and the blood loss was 964 g. No complications occurred during the perioperative period. The local kyphosis angle improved from 29° before surgery to −12° immediately after surgery and to −7° at the final follow-up. There was no cage subsidence and bone fusion was observed at one year after the surgery (Figure 4).

## 5. Discussion

As the major etiology of OVF is the collapse of the anterior column of the vertebral body, anterior vertebral reconstruction, which was first reported by Kaneda in 1992, is one of the most common surgical methods [9]. The thoracolumbar spine, the most common site of OVF, has a high anterior load-bearing capacity and therefore the reconstruction of the anterior column is essential from a biomechanical point of view. However, in patients with multilevel vertebral fractures due to severe osteoporosis, additional posterior fixation has been recommended. Furthermore, the correction of rigid kyphotic deformities has been reported to be difficult to accomplish by ASF alone [11,16]. In those cases, three-column osteotomy might be required as this method has the advantage in correcting rigid kyphotic deformities, although this surgery is associated with a high rate of perioperative complications such as massive bleeding [15,19,20].

Matsuyama et al. reported on posterior fusion combined with vertebroplasty as a new minimally invasive surgery for pseudoarthrosis after OVF [12]. This technique is one of the most frequently performed methods for OVF in Japan. As the anterior elements can be reconstructed easily from the posterior approach alone, there is only a low risk of potential injuries to thoracic and abdominal organs or the major vessels which may occur with the anterior approach, which the majority of spine surgeons are less familiar with. One of the disadvantages of posterior surgery without rigid anterior reconstruction is the high incidence of correction loss [15]. On the other hand, Uchida et al. reported that the correction loss angle was almost the same when anterior spinal fusion alone was compared with posterior fusion combined with vertebroplasty, so the efficacy of anterior reconstruction by vertebroplasty has still not been fully demonstrated [14]. Sudo et al. reported that the use of an ultra-high molecular weight polyethylene cable system was effective in reducing the correction loss angle [13]. Conventional anteroposterior combined surgery was also reported to be effective in reducing the correction loss angle, although this method is relatively highly invasive [23].

As MIST surgery has recently been more widely performed, vertebral corpectomy with a minimally invasive lateral approach similar to L-LIF is widely used, making it possible to perform combined anteroposterior VBR in a less invasive procedure than before [26,28,29,30]. Corpectomy with a minimally invasive lateral approach can achieve a fixation with less adjacent tissue damage. Moreover, the proper use of these methods has been shown to shorten recovery times, as well as to reduce blood loss and perioperative complications [26].

There are many different cages available for VBR, and one of the most useful cages is an expandable circular footplate cage. However, smaller circular footplate designs have been associated with subsidence and correction loss. Cages with wide, rectangular footplates have been shown to reduce subsidence by settling down on the apophyseal ring, which has been known to possess the strongest and most dense vertebral endplate bone [31]. In a mechanical verification using a cadaver to compare rectangular and circular footplate cages, the rectangular footplate cage significantly reduced intervertebral motions and the cage subsidence was less [31,32]. As patients with OVF have vertebral fragility, it may be difficult to stabilize the fractured vertebra with a circular plate. VBR combined with an expandable cage with a wide footplate using a minimally invasive lateral approach may be one of the most suitable options for elderly OVF patients with severe osteoporosis. However, as this technique is relatively new, there is a lack of evidence to determine the optimal surgical procedure. Therefore, we investigated the surgical results of this technique to verify the most appropriate procedure.

Taiji et al. reported a result of a similar procedure in 16 OVF patients and reported that the average correction loss angle was 8.5° with one above–one below fusion [24]. The average correction loss angle in our study was 12.3° for less than two above–two below fixation, which was larger than the study by Taiji. In their study, lamina hooks were used combined with pedicle screws, and in the patients whose kyphosis was unable to be corrected at prone position, anterior release was initially performed followed by P-A surgery, which required the patient’s position to be changed twice during surgery. These differences may be the reasons for the correction loss angle being smaller even in the one above–one below fixation in their study. Our results indicate that at least two above–two below fixation is necessary to minimize the correction loss angle and to prevent cage subsidence, which can be achieved with only a single position change during surgery. To fuse an even longer range may have advantages in terms of correction loss and cage subsidence prevention, but AVF and proximal junctional fracture (PJF) have been reported to increase in these long-range fixations [33]. In fact, there were more UIV or LIV fractures and AVF in 5VG than in 4LVG, although the incidence was not significantly different in our study (Table 3). Therefore, patients with kyphotic deformities after OVF requiring long-range fixation from the lower thoracic spine to the ilium should be monitored carefully postoperatively for a longer period for these mechanical complications. UIV or LIV fractures were the most common mechanical failures in this study. This complication is considered as multifactorial, such as global alignment, bone mineral density (BMD), and affected vertebra level. As not all the patients were able to evaluate preoperative global alignment, the etiology of junctional fractures was not able to be properly evaluated in this study.

Our comparison study between 5VG and 4LVG revealed significant differences in gender, with the follow-up period and surgical sequence with 4LVG having more males and longer follow-up with more A-P surgery (Table 3). This might be due to the fact that males were considered to have better bone quality than females, and therefore, the shorter range of fixation was used in males. Moreover, one above–one below fixation with the A-P sequence was used in the early phase of VBR introduction in our case series, but this was eventually changed to at least two above–two below fixation with the surgical sequence, as in the above-described method.

There was a tendency for more women with postoperative residual kyphosis in the comparison between the satisfactory correction and inadequate correction. It is possible that the women had poorer bone quality resulting in poor correction with residual kyphosis. However, there was no significant association between residual local kyphosis immediately after surgery and implant failure such as cage subsidence. This may be due to the various locations of the affected vertebra, the lumbar spine having a lordotic alignment, and the thoracic spine having a kyphotic alignment. Therefore, the influence of the residual kyphosis may differ according to the level of OVF. Another reason might be due to the lack of global alignment assessment as many patients had difficulty in keeping the standing position in the preoperative period. Terai et al. evaluated the pre- and post-operative global alignment of 54 patients with pseudoarthrosis after OVF who underwent VBR followed by posterior fixation with PPS. They compared the poor global alignment group with a pre-operative sagittal vertical axis (SVA) > 95 mm to the good global alignment group with SVA ≤ 95 mm, which showed a higher incidence of AVF and a high correction loss angle in the poor global alignment group [25]. This indicates that global alignment rather than local kyphosis correction may have a higher association with implant failure.

In our study, three patients developed endplate injury and three others developed cage floatation. Five of these patients had A-P surgery, and three of them had preoperative bone bridging with adjacent vertebrae. In the patients with rigid kyphosis due to bone bridging with adjacent vertebrae, anterior release was required as the first surgery for kyphosis correction. However, even if the intervertebral space was opened by anterior release, endplate injury may have occurred by inserting the cage into an insufficient space. Furthermore, even if the cage had been properly fitted by anterior surgery, cage floatation may have occurred in those patients who required long-range posterior fixation. The reason for cage floatation is that posterior surgery with the cantilever technique has a strong corrective force which is difficult to control, so it may create a gap between the endplate and the cage even though the cage has been properly placed during the first surgery. In these cases, three-stage surgery, which is the posterior corrective fusion after anterior release followed by VBR, might be the solution to prevent endplate injury or cage floatation, although this is undesirable for elderly patients due to high surgical invasion. Three-column osteotomy such as pedicle subtraction osteotomy (PSO) or posterior vertebral column resection (PVCR) should be considered for those patients who may not benefit from combined anteroposterior surgery [15,19,20]. However, these surgeries pose a higher risk, such as massive blood loss to elderly patients who tend to have an insufficient reserve. An appropriate surgical strategy for these patients should be the focus of future studies.

The limitations of this study concern the small number of patients and the different fixation ranges. Furthermore, neither the global alignment nor the objective clinical outcomes were evaluated. However, since this surgical technique is relatively new, we aimed to evaluate pitfalls or perioperative complications of this technique and to share our experiences with spine surgeons who are involved in the surgical treatment for OVF. Therefore, we included the patients with a minimum of one-year follow-up.

## 6. Conclusions

This study investigated the radiological results of 38 patients who had combined anteroposterior fusion with VBR using a minimally invasive lateral approach to reveal the optimal surgical procedure for OVF. Our results showed that at least two above–two below fixation was preferred to minimize the correction loss angle and cage subsidence with a single position change during surgery. In terms of surgical sequence, we recommend P-A surgery for patients whose kyphosis is flexible in a supine position, whereas A-P surgery should be performed in those patients with rigid kyphosis due to anterior bony fusion. Furthermore, careful attention should be paid to the endplate injury or cage floatation in A-P surgery, especially for those with insufficient intervertebral space or those undergoing long-range posterior fixation. The optimal surgical technique for such patients is a subject for further study.

## Figures and Tables

**Figure 1 jcm-11-00629-f001:**
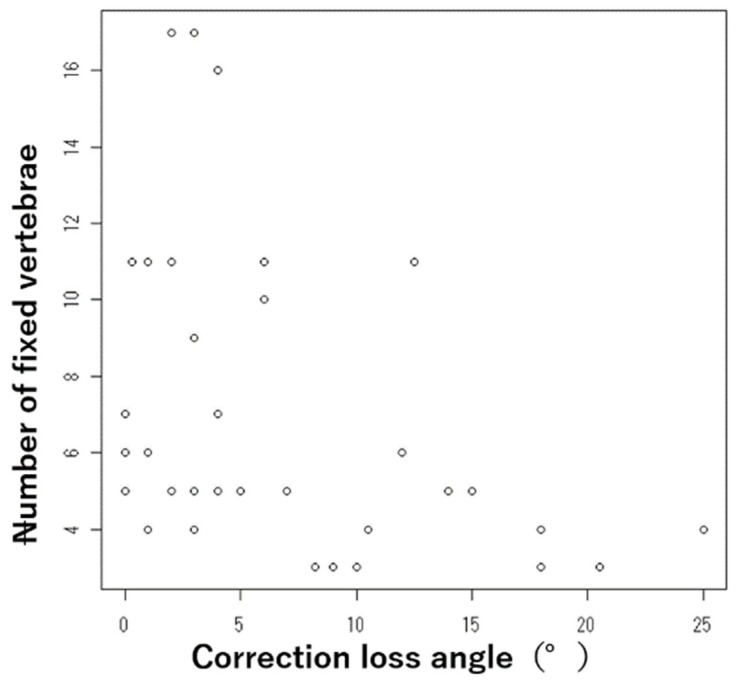
Relationship between the number of fixed vertebrae and the correction loss angle.

**Figure 2 jcm-11-00629-f002:**
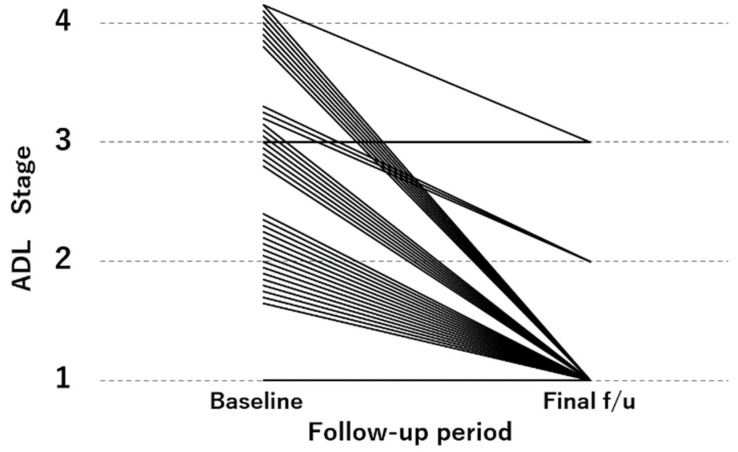
ADL at the baseline and at the final follow-up. Final f/u: final follow-up.

**Figure 3 jcm-11-00629-f003:**
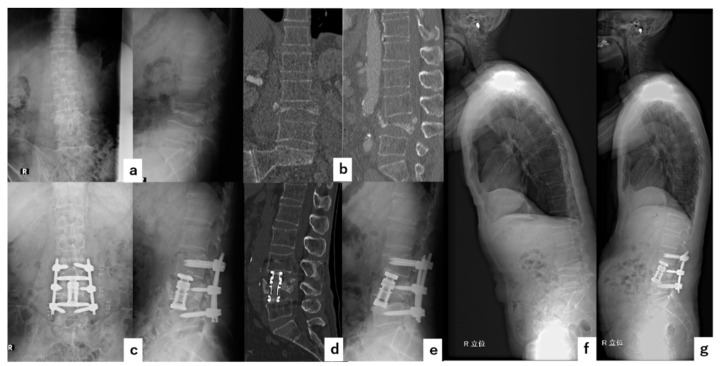
Illustrative Case 1. A sixty-nine-year-old male underwent a combined anteroposterior short-segment fixation with a wide footplate expandable cage for pseudoarthrosis after OVF at L4. (**a**) Preoperative anteroposterior and lateral radiographs showing that the local kyphotic angle was 24°. (**b**) Preoperative coronal and sagittal CT. (**c**) Immediate postoperative anteroposterior and lateral radiographs. (**d**) Postoperative sagittal CT. (**e**) Final follow-up lateral radiograph showing that the cage subsidence and local kyphosis angle was −11°. (**f**) Preoperative whole-spine lateral radiograph. (**g**) Postoperative whole-spine lateral radiograph.

**Figure 4 jcm-11-00629-f004:**
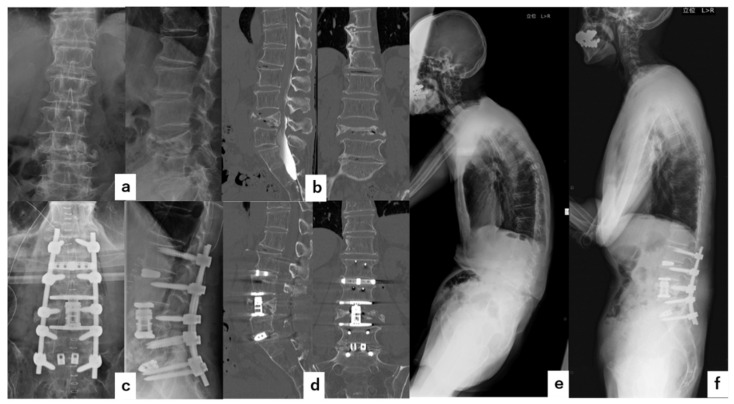
Illustrative Case 2. A seventy-five-year-old male underwent a combined anteroposterior two above–two below fixation with a wide footplate expandable cage for pseudarthrosis after OVF at L4. (**a**) Preoperative anteroposterior and lateral radiographs showing that the local kyphosis angle was 29°. (**b**) Preoperative coronal and sagittal CT. (**c**) Immediate postoperative anteroposterior and lateral radiographs. (**d**) Postoperative sagittal CT. (**e**) Preoperative whole-spine lateral radiograph. (**f**) Postoperative whole-spine lateral radiograph.

**Table 1 jcm-11-00629-t001:** Demographic data of patients and surgical outcomes.

Variables		Value
Number of patients		38
Age		75.2 ± 7.4 (53–85)
Gender (male/female)		17/21
Follow-up period (months)		21.2 ± 10.4 (12–51)
Affected vertebra	Thoracolumbar junction level (T11-L2)	27 (71.1%)
	Lumbar level (L3-4)	11 (28.9%)
Sequence of surgery	A-P	21 (55.3%)
	P-A	17 (44.7%)
Number of fixed vertebrae		6.9 ± 3.9 (3–17)
Local kyphotic angle (°)	Pre-operative angle	29.3 ± 17.4(−21–62)
	Correction loss angle	4.9 ± 8.4 (−15–20)
Mechanical failure (%)	UIV/LIV fracture	12 (31.6%)
	Cage subsidence	10 (26.3%)
	PS loosening	8 (21.1%)
	Cage floatation	3 (7.9%)
	Endplate injury	3 (7.9%)
	Adjacent vertebral fracture	2 (5.3%)

The values are given as mean value +/− standard deviation, and the range in (−). A-P: anterior surgery followed by posterior surgery, P-A: posterior surgery followed by anterior surgery, UIV: upper instrumented vertebra, LIV: lower instrumented vertebra, PS: pedicle screw.

**Table 2 jcm-11-00629-t002:** Comparison between patients with cage subsidence (+) and without cage subsidence (−).

Variables		Cage Subsidence (+)	Cage Subsidence (−)	*p*-Value
Number of patients		10	28	
Age		72.4 ± 8.5	76.1 ± 7.1	0.118
Gender (male/female)		6/4	11/17	0.223
Follow-up period (months)		25.8 ± 10.7	19.6 ± 10.1	0.083
Sequence of surgery (%)	A-P	8 (80%)	13 (46.4%)	0.069
	P-A	2 (20%)	15 (53.6%)	
Number of fixed vertebrae		5.4 ± 4.5	7.4 ± 3.6	0.003

The values are given as mean value +/− standard deviation. A-P: anterior surgery followed by posterior surgery, P-A: posterior surgery followed by anterior surgery.

**Table 3 jcm-11-00629-t003:** Comparison between the five vertebrae fixation group (5VG) and the four or less vertebrae fixation group (4LVG).

Variables		5VG	4LVG	*p*-Value
Number of patients		12	10	
Age		75.6 ± 8.3	72.9 ± 9.8	0.39
Gender (male/female)		2/10	6/4	0.048
Follow-up periods (months)		15.8 ± 5.0	25.1 ± 15.8	0.017
Sequence of surgery (%)	A-P	3 (25%)	7 (70%)	0.046
	P-A	9 (75%)	3 (30%)	
Correction loss angle (°)		4.8 ± 5.0	12.3 ± 7.8	0.016
Cage subsidence (%)		0 (0%)	8 (80%)	0.0001
UIV/LIV fracture (%)		8 (66.7%)	4(40%)	0.21
Adjacent vertebral fracture (%)		1 (8.3%)	0 (0%)	0.55

The values are given as mean value +/− standard deviation. A-P: anterior surgery followed by posterior surgery, P-A: posterior surgery followed by anterior surgery.

**Table 4 jcm-11-00629-t004:** Comparison between patients with a postoperative local kyphotic angle of less than 5 degrees and those with 5 degrees or more in the 5 vertebrae fixation group (5VG) and the 4 or less vertebrae fixation group (4LVG).

Variables		Post-Operative Local Kyphosis <5°	Post-Operative Local Kyphosis ≥5	*p*-Value
Number of patients		13	9	
Age		77.2 ± 5.8	73.1 ± 9.7	0.226
Gender (male/female)		7/6	1/8	0.052
Follow-up periods (months)		19.4 ± 8.1	21.0 ± 9.2	0.784
Sequence of surgery (%)	A-P	6 (46.2%)	4 (44.4%)	0.639
	P-A	7 (53.8%)	5 (55.6%)	
Correction loss angle (°)		7.0 ± 7.2	10.1 ± 7.4	0.299
Cage subsidence (%)		4 (30.8%)	4 (44.4%)	0.246

The values are given as mean value +/− standard deviation. A-P: Anterior surgery followed by posterior surgery. P-A: Posterior surgery followed by anterior surgery.

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
