# Peer review of "Radiological Evaluation of Combined Anteroposterior Fusion with Vertebral Body Replacement Using a Minimally Invasive Lateral Approach for Osteoporotic Vertebral Fractures: Verification of Optimal Surgical Procedure"

_jcm, 2022, doi:10.3390/jcm11030629_

Round 1

Reviewer 1 Report

The manuscript “Radiological evaluation of combined anteroposterior fusion with vertebral body replacement using a minimally invasive lateral approach for osteoporotic vertebral fractures.-Verification of optimal surgical procedure” Takumi Takeuchi, Kenichiro Yamagishi, Kazumasa Konishi, Hideto Sano, Masato Takahashi, Shoichi Ichimura, Hitoshi Kono, Masaichi Hasegawa, Naobumi Hosogane aimed to evaluate the radiological results of combined anteroposterior fusion using VBR and to investigate its optimal surgical procedure.

Below are my comments and remarks regarding the article:

1. Definition of bone fusion was not given
2. Please provide the instrumentation used (type, manufacturer)
3. Who performed the surgical procedures and what was their experience
4. Please convert the tables to text
5.  What is the explanation  for  common male in the 4LVG group ?
6. What is the explanation for the postoperative local kyphotic angle> 5 was more common in female?

Reviewer 2 Report

Well written study, examining a difficult problem in the aging population that sustains a OVF.  A few questions and comments:

  1. While you demonstrated healing in all of your cases, is less than 2 year avg follow up sufficient to answer the question of success for this procedure? Particularly relevant for those with longer constructs.
  2. While the TL junction is the most common site for OVF, both case examples are lumbar reconstructions.  It would be helpful for the paper to delineate the distribution of location of OVF in your series, i.e. 30% lumbar (levels) and 70% TL (L1, T12 etc).  Any OVF at L5, and or can MIST be performed at this level?
    1. it would be helpful to clarify what is fundamentally different about your MIST anterior approach versus an 'open' anterior retro-peritoneal approach thru a small incision.
  3. I think value would be added for the results to reflect global spinal alignment parameters in the series of patients, as opposed to just local alignment, since this may be a factor for later complications, as you point out in the discussion section.  Was this looked at?  This would also have bearing on addressing the disparity between a procedure that has an overall complication rate of 68% yet a healing rate of 100%. In other words, e.g., they all healed and global alignment was satisfactory, in spite of  high mechanical failure...
  4. The most common mechanical failure was UIV/LIV fracture, yet most attention was spent on cage subsidence.  Any thoughts on this problem with regard to etiology - perhaps global alignment issues are related, or should VP be added at upper level of fixation or the level above?
  5. Please clarify the section lines 139-141 discussing the average LKA pre and post op and the mean loss of correction as it is not clear how you arrived at a mean loss of 6.3 degrees.
  6. in fig 3 and 4 there are two different font sizes
